# Identification of Gate Turn-Off Thyristor Switching Patterns Using Acoustic Emission Sensors

**DOI:** 10.3390/s21010070

**Published:** 2020-12-24

**Authors:** Maciej Kozak, Artur Bejger, Arkadiusz Tomczak

**Affiliations:** 1Faculty of Mechatronics and Electrical Engineering, Maritime University of Szczecin, Wały Chrobrego 1-2, 70-500 Szczecin, Poland; 2Faculty of Mechanical Engineering, Maritime University of Szczecin, Wały Chrobrego 1-2, 70-500 Szczecin, Poland; a.bejger@am.szczecin.pl; 3Faculty of Navigation, Maritime University of Szczecin, Wały Chrobrego 1-2, 70-500 Szczecin, Poland; a.tomczak@am.szczecin.pl

**Keywords:** acoustic emission, sensor, transducer, gate turn-off thyristor, power electronics

## Abstract

Modern seagoing ships are often equipped with converters which utilize semiconductor power electronics devices like thyristors or power transistors. Most of them are used in driving applications such as powerful main propulsion plants, auxiliary podded drives and thrusters. When it comes to main propulsion drives the power gets seriously high, thus the need for use of medium voltage power electronics devices arises. As it turns out, power electronic parts are the most susceptible to faults or failures in the whole electric drive system. These devices require efficient cooling, so manufacturers design housings in a way that best dissipates heat from the inside of the chips to the metal housing. This results in susceptibility to damage due to the heterogeneity of combined materials and the difference in temperature expansion of elements inside the power device. Currently used methods of prediction of damage and wear of semiconductor elements are limited to measurements of electrical quantities generated by devices during operation and not quite effective in case of early-stage damage to semiconductor layers. The article presents an introduction and preliminary tests of a method utilizing an acoustic emission sensor which can be used in detecting early stage damages of the gate turn-off thyristor. Theoretical considerations and chosen experimental results of initial measurements of acoustic emission signals of the medium voltage gate turn-off thyristor are presented.

## 1. Introduction to the Subject Matter

The analysis of acoustic emission (AE) signals is widely used to detect damage in solid materials. The most popular areas of application are the supervision of fatigue phenomena and cracks in steel structures [1,2,3,4], rolling elements in bearings [5,6] or the occurrence of partial discharges in power transformers [7,8] and medium voltage switchgears [9,10,11]. There have been attempts at recognizing AE signals in low voltage insulated gate bipolar transistors (IGBTs) [12,13,14,15,16] even with changes of junction and case temperature [17]. It must be noted that AE tests were applied to small packaging low-voltage semiconductors (without electrical insulation inserted between AE sensor and device case) but it can be assumed that the propagation of elastic waves in much bigger structures will behave differently because of the extended internal volume of different types of packaging and the use of insulating inserts for medium-voltage operation.

That is the main reason gate turn-off (GTO) thyristors and newer gate-controlled thyristors are still widely used in high-power medium voltage applications such as ships main propulsion plants. The article presents possibility of using an acoustic emission sensor for diagnostics of early-stage dislocations and structural cracks inside gate turn-off thyristors.

### 1.1. Types of Power Electronics Devices Cases and Packaging

The silicon-controlled rectifier thyristor (SCR) can be considered as the first semi-controlled device that started medium voltage power electronics. Currently, this device in its basic not fully controlled form has largely disappeared from medium voltage applications, having gradually been replaced by exclusive GTOs and for several years now, by their upgraded version called gate controlled thyristors (GCTs) and integrated gate controlled thyristors (IGCTs). It can be assumed that the integrated gated controlled thyristor consists of the GTO structure integrated with an electronic driver circuit providing operation within a safe operation range, reduced switching losses and a short storage time [18].

Thyristors designed for medium voltage operation are usually made as a ceramic disc with metal anode and cathode plates. For proper operation they have to be efficiently cooled with means of a radiator stuck to the medium voltage plates. This is of course a main source of noises coming out of strong electric field (because of vicinity of high voltage electrodes) and electromagnetic field (coming from the current flow). Due to the switching cycle, the internal structure and inner silicon layers will be subjected to continuous mechanical stresses resulting from the repeated heating and cooling cycles.

Repeating temperature changes are the main cause of thermal stress, wear and in the end semiconductor structure failures. For designers these stresses are a quite bit of concern in the design and operation of power electronics devices, and a large number of publications have been devoted to thermal phenomena occurring inside of silicone-based electronics.

It can be assumed that the power electronics devices thermomechanical design problems can be defined in terms of the following categories [19]:-slight temperature changes causing thermoelastic deformation,-stress fields resulting from major changes of junction and internal temperature and displacements, along with erratic temperature distribution effects.-elastic or elastoplastic deformation due to the effects such as thermal shock, creep, stress relaxation, stress rupture, and thermal fatigue.

To master some of aforementioned issues two main technologies are utilized in the manufacture of power thyristors: alloying and free-floating silicon technology [20].

Vacuum brazing using aluminum and silicon alloys is a commonly used method in alloy brazing to join silicon chips with molybdenum thermal compensators. Use of such technology provides a firm silicon chip-molybdenum disc junction with good cycling capacity and quite low thermal impedance.

In this case some external force is needed for installation purposes from the cathode side of thyristor is required to prove firm thermal contact. Even so, since alloying is a high-temperature process, thermomechanical stresses show up in Si-Mo structure because of the different thermal expansion coefficients of silicon and molybdenum.

However, when combining silicone chips with larger diameter outer chassis plates, this issue becomes even more important. Free-floating silicon technology introduces a semiconductor layer with the cathode and anode, metallization between them and thermal compensators. Because of the lack of soldered joints, only pressure thermal and electrical contacts between the silicon plate and thermal compensator can be distinguished. The advantage of the pressure contact design is the absence of deformation and residual stresses that occur when soldering a silicon plate with a thermal expansion joint due to the difference between the expansion coefficients. This feature is extremely important in the fabrication of semiconductor components, especially those of bigger diameter. Another important advantage of free-flowing silicon technology is that the surface layers of silicon do not dissolve during the manufacturing process. On the other hand, there is higher thermal resistance from the anode side in comparison to the soldering technology.

Another factor related to temperature expansion phenomena and changes in geometric dimensions is the type of thyristor housing [21].

There are several types of thyristor packages presented in Figure 1, ranging from low power devices enclosed in small plastic housings, then bolted medium power devices to flat-pack (or press-pack) systems for high power and high voltage systems. Due to technological restrictions the thyristors enclosed in the flat-pack cases must be mounted under certain and precisely controlled pressure in order to get proper electrical and thermal contact between the semiconductor layer and the external metal electrodes. Huge diameter thyristors should not be directly soldered or glued to the large copper pole piece of the flat-pack because of the significant difference in the coefficient of thermal expansion (CTE). To avoid this issue the contact for both anode and cathode is obtained by means of pressure assembly.

Due to the differences in the way the inner layers are installed and connected, as well as the type of enclosure, it is expected that the acoustic emission signals will also vary during switching.

### 1.2. Phenomena inside of Cycling GTO Thyristor

Gate turn-off thyristors, as opposed to SCRs are fully controllable switches that allow turning on and off by applying a voltage to the gate lead. Classic silicon rectified thyristors can only be switched off by decreasing the anode current below the value of the holding current. Therefore, semi- controlled SCR thyristors are not the best choice for direct current applications. The GTO thyristor can be turned on by a certain current injected into the gate and anode gate and it conducts until this sustaining current has a proper value. It is possible to turn off this kind of thyristor simply by applying a gate current signal of negative polarity. The turn-on phenomenon in GTOs is more reliable than in a SCR thyristor and a continuous gate-anode current should be maintained in order to improve reliability.

An important issue in constructing power electronic devices is to minimize stray inductance inside the structure and to reduce the inductance value between terminals and anode and cathode plates. The mounting to the terminals should be designed in the way it minimizes inductances in order to eliminate overvoltage spikes during the switching process. Due to parasitic inductances inside the semiconductor structure, there may occur high frequency ringing when changing states fast. As occurs in every power electronics design, the capacitance of semiconductor layers and cast also creates the problem of a mutual electromagnetic interference noise between two or more devices when placed close to each other. In some situations, these capacitances may cause firing of gate circuitry charging and thus unwanted switching and serious equipment failures [21].

In addition to the aforementioned, designers of power electronic devices must primarily take into account the thermal effects occurring during conduction and switching of thyristors. In contrast to SCR thyristors, during the turn-on time the GTO thyristor needs a gate current ranging from 20 up to 30 percent of the conductive current at all times. This additionally increases the temperature of the junctions and greatly affects the deformations and dislocations within the structure. The turn-off process is initiated by applying a negative voltage between the gate and cathode of thyristor. The forward current is used to induce a cathode-gate voltage what results in a decrease of the anode forward current, and thyristor will switch off. As for the inner semiconductor structure there are identical in width and length tiny emitter mesa spots distributed inside the structure (Figure 2), which allow similar flow of the turn-off current in every path.

In the case of no homogeneous current flow during the turn-off period can result in filamentation and with it in dynamic avalanche thus enormous silicon element heating and burn out.

The fatigue of semiconducting material comes from coefficient of thermal expansion mismatch between sticking materials of different coefficients because the switching frequency thermal cycling can significantly increase the fatigue process what can be detected with means of elastic waves inside the structure. With the temperature changes and variations, the internal wafers expand and contract at different rates what is main cause for soldered layer cracking and debonding. Some CTEs of popular materials used in power electronics devices are given in the Table 1. The extensive temperature rise also plays an important role in the chemical degradation processes such as dendritic growth and protrusion migration, therefore keeping the size of cooling plates large enough and ensuring efficient cooling is one of the most important issues related to the design and proper operation of power electronic devices [22].

The time rates of anode current during turn-on and anode-cathode voltage while turn-off are parameters needed to properly control the system and achieve reliable operation. These values should never exceed the permissible level stated by the manufacturer. When in the conducting state the areas of a device near the gate begin to conduct current sufficient time must be provided for the whole cathode area to begin conducting before the short-circuit currents become too high. If the rate of rise of forward anode-cathode voltage is too high thyristors can trigger (or self-trigger) into a conduction mode from a forward-blocking mode because of junction capacitance. To avoid any influence of switching voltage spikes additional capacitors and resistors known as snubbers are used as protection circuits. These can be found in other power semiconductor devices. A lack of snubbers can damage the silicon structure and contribute to the creation of unwanted acoustic emission signals which distort the frequency spectrum. All of aforementioned processes are the part of the thyristor wafer wear process along with natural structure aging. According to [23] due to aging some subtle changes occur such as waveform anode voltage alterations during turn on processes indicating physical changes in the thyristor gating circuit.

### 1.3. Acoustic Emission Signals and Analysis

Because of the cyclic nature of GTO switching the phenomena occurring inside its structure will be most notable at the working frequencies of the thyristor. There can be distinguished the following types of thyristor operation: the turning on, conducting/blocking state and turning-off events. This article covers use of AE signals obtained in the rectifying mode of operation so the expected elastic waves spreading across press-pack GTO structure would eventually contain power grid frequencies and its multiples as some of internal wave bouncing occurs. The acoustic signal analysis can be recognized as mathematical methods of signal processing in order to obtain valuable information about the inner state of a solid object. In the case of power electronics devices, the signals coming from a sensor can include a lot of “contamination” in a form of EMI or strong electric field noise which must be removed. The useful signals obtained during nominal parameter (current, voltage or junction temperature) operation can become a pattern and any abnormal state of the inner structure of the thyristor should result in change of the AE signal. When analyzing AE signals in the time domain, namely acceleration amplitude against time to quantify the strength of an elastic wave signal, a few parameters are needed (and observable): amplitude, peak-to-peak value and RMS. As was mentioned any signal coming from a semiconductor device includes a lot of additional information which can be represented as a mix of signals of different amplitudes and frequencies. The analysis of such in the time domain is not very useful so proper methods of signal analysis were introduced and now are widely used in diagnostics. Each of these have different properties and better fit various applications. These are the fast Fourier transform (FFT), frequency spectrogram and power spectral density (PSD).

The FFT decomposes the obtained signal into a Fourier series containing individual sine wave components. In numerical-based applications the fast Fourier transform in its fastest Radix-2 decimation-in-time (DIT) form is willingly used. It can be easily applied to the digital signal processor code [24] and operate in real-time. The result of Radix-2 operation over incoming signals is data containing acceleration amplitude as a function of frequency. This data enables signals analysis in the frequency domain and in diagnostic applications, the vast majority of analyses are typically done in such a domain. The FFT is fairly good for the detection and analysis of stationary state signals but with prolonged operation of the power electronics devices, the parameters like junction temperature along with geometrical dimensions (due to CTE) will change. This in turn would have an impact on the acoustic emission signals coming from the structure. In such a case it is more convenient to use the frequency spectrogram which basically creates and combines a series of FFTs and overlaps them into one plot to illustrate how the spectrum in the frequency domain changes with time. The frequency spectrogram can be very useful to illustrate how the spectrum of the acoustic emission varies in a changing environment. Another way of describing the contribution of individual frequency components to the total signal is the so-called power spectral density. A lot of acoustic signals during transient states include some noise arising from states that are dynamically changing at many frequencies at the same time. While the FFT is good enough at analyzing elastic waves signals when there is a finite number of dominant frequency components the power spectral densities are mainly used to characterize random signals. Because an ideal spectrum consists of an infinite number of components, easy calculation of the total dissipated power is not possible, so it is more convenient to denote it by power per frequency (or bandwidth) obtaining the units of V^2^/Hz. Such a spectrum is called power density spectrum (PDS) and the value of the density is called power spectral density (PSD) [25]. The values of PSD are obtained by multiplying each frequency bin in a fast Fourier transform by its complex conjugate which results in the real spectrum of amplitude described in g^2^. The power spectral density analysis in the case of changing, noisy signals seems to be more useful than a FFT because amplitude value is normalized to the frequency bin width which in turn leads to units described as g^2^/Hz. Use of PSD has another advantage over FFT namely the dependency on bins width disappears, so comparison signals of different lengths become straightforward.

Considering its Fourier transform over the interval ±T/2 as X(ω) for a time domain signal denoted as *x*(*t*) the power spectral density is given by:(1)$$S_{X}(\omega )=\underset{T\to \infty }{lim}\frac{E\{{\left|X(\omega )\right|}^{2}\}}{T}$$
and the area under the spectrum curve represents the total power of the signal, which is given by following equation:
(2)$$x^{-2}=\overset{+\infty }{\underset{-\infty }{\int }}S_{X}\left(f\right)df=2\int_{0}^{\infty }S_{X}\left(f\right)df$$

What can be considered as an advantage is that nonstationary AE signal time series analysis presents how the energy is distributed during a measurement time span. Nowadays there is widely used software which performs the aforementioned operations in real-time thus the analysis and on-line monitoring of internal state of materials is also possible for offline work. There are of course preprogrammed, compact systems created for such operations, but they are dedicated to specialized areas and the signals coming out can be heavily filtered what makes them not very practical for power electronics emission signals analysis [26].

## 2. Materials and Methods Used in the Experiments

In practical applications of acoustic emission sensors, they are used to detect the high frequency energy signals which are generated in inner structure cracks when part of the material is displaced or when the contacting layers have different expansion coefficients. These signals are spread in all directions inside the structure and of course they bend and bounce at the material borders of different densities. Knowing these issues there is a good chance to measure such signals (appearing as elastic waves), convert them into electrical signals and send them to a monitoring or diagnostic system. The more complex the structure is the more bounced, bent and overlaid signals must be expected but after precise filtering a lot of diagnostic information can still be obtained. The amplitude of electrical signals is quite low (up to dozens of millivolts) so the signals are amplified, and final step detectors have built-in filters. These signals can be analyzed in different ways depending on the nature of the expected phenomena. In most of the cases such signals are used for the detection of cracks, breaks and wear inside mechanical structures and the detection of electrical partial discharges. All of the aforementioned facts are well known and the detection devices have filters which are tuned in order to amplify interesting and well-known frequencies indicating the beginning of destructive processes. The nature of phenomena occurring inside multilayered press-pack type semiconductor devices is not entirely known, so it is crucial to get AE signals unaltered in any way which means that widely used front end commercial and dedicated recorders and detecting devices cannot be used. The most convenient and obvious way to get the wide spectrum of acoustic emission signals generated inside the semiconductor structure is to use a wideband AE sensor connected directly to an oscilloscope without preamplifiers and filters on. This allows the observation of pure, raw electrical signals which of course include a lot of useless information, but further analysis can reveal interesting behavior of the switching elements and can lead to the creation of unique, proper switch pattern signals. This pattern can be used for real-time observation of outgoing signals and can report early-stage malfunctions or premature failures.

As long as there are no references to informative sources covering frequencies generated inside switching thyristors the wide band frequency sensor was typed for use and detection of every kind of acoustic emission occurring in the tested semiconductor structure.

The thin disc-shaped piezoelectric material which converts material deformation into electrical signal is an important, active element of an AE sensor. To assure good electrical conductivity the piezoelectric surfaces are metalized and in order to prevent EMI interferences the whole structure is placed inside a closed cylinder made out of metal. Titanate and zirconate crystals mixed with other materials are widely used in AE transducers. The piezoelectric properties are obtained by ceramic material poling. This process involves heating the element above the Curie point in the presence of electric field and finally it produces asymmetrical internal crystal structure [27].

As the most promising and widely used device the WSα factory-calibrated sensor was chosen. The frequencies detected by WSα according to Figure 3 ranging from 0 up to 1000 kHz, thus any potentially interesting frequencies are fully covered by the chosen sensor.

### 2.1. Recognition of AE Sensor Immunity to EMI Noise

The major question is what kind of signals an acoustic emission sensor detects while operating at a close distance to the conducting current object. In the article the high-power GTO thyristor is an object which produces specific AE raw signals in blocking/conducting mode and a notably high current flows through its structure. This high current produces a magnetic field which moves through the semiconducting layers, metal case and wiring harness so it will have an impact on transducer operation and final readings. Because the propagation paths and nature of AE signals in semiconductor layers is not exactly known, it seems to be reasonable to perform measurements with use of wide-band sensor and apply filtering at the very end.

The construction of the wideband AE sensor (WSα) used in our experiments consists of layers which are placed on the ceramic plate. This ceramic plate is laid on the surface of the tested material or semiconductor device (see Figure 4). In addition, it creates an insulating layer to prevent short circuits on the conducting surface.

Inside the metal case there is a completely enclosed crystal for RFI/EMI immunity, which converts the vibrations into electrical signals. Because of the ceramic plate width and the presence of a strong magnetic field created by the current flowing through the tested thyristor there arises questions about the magnetic field influence on the measurements. It is crucial to know how prone the sensor is to EMI noise before conducting further acoustic emission measurements. Unfortunately, acoustic emission sensor manufacturers do not provide detailed information on their resistance to magnetic and electric field interference, so it was necessary to check sensor responses in the presence of a magnetic field. To answer this question, a laboratory stand was set up, which was equipped with an autotransformer, a laboratory coil, diode, acoustic emission and magnetic field Hall-effect sensor. The latter transducer used was an AH49E type with a LM393 amplifier embedded on a PCB and powered by a battery. Both sensors were placed on top of the coil secured in the place and with means of an autotransformer the current flow across the coil was changed (Figure 5). The tests covered placing the AE sensor away from top of the solenoid just to find relationship between the near magnetic field and the acoustic emission transducer readings. The AE sensor was placed on a paper stack of different thicknesses of 0.05 mm (one sheet), 20 mm, 45 mm and 85 mm.

The sinusoidal voltage coming from autotransformer was applied to the coil and acoustic emission, magnetic field and current signals were recorded on the oscilloscope.

Taking into account that the coil wire length is much larger than its diameter the magnetic field within the winding is given by *B* = *μ*_0_*nl* where *μ*_0_ is the permeability constant, n—number of coil turns and *I* denotes current flow. With a current flowing through the coils, the magnetic field produced within the solenoid can be written as:(3)$$B=\mu_{0}\left(\frac{N}{l}\right)I$$

Knowing that the magnetic flux for a given area equals to the area value multiplied by the component of magnetic field perpendicular to the penetrated area according to following equation.

This relationship can be presented as:(4)$$\Phi_{m}=BS=\frac{\mu_{0}NS}{l}I$$

Assuming that magnetic flux depends on current flow and introducing parameter of the coil known as self-inductance $\Phi_{m}=L\;I$ the inductance of N turns inductor can be expressed as:(5)$$L_{coil}=\frac{N\Phi_{m}}{I}=\frac{\mu_{0}N^{2}S}{l}$$
where *l* is the coil wire length, *S* means cross-sectional area and *N* is a number of copper wire turns.

After applying known value of the current and coil parameters into Equations (4) and (5), the values of magnetic induction and flux were calculated. These figures have been confirmed by the results of measurement tests using a Hall sensor attached to the scope.

The solenoid chosen for the test had a self-inductance of 0.17 H with diameter equal to 11 cm, and the current flowing across coil was set to 1 A RMS (1.41 A in peak). The value of magnetic field measured directly in the center point of the coil was equal to 25 Gauss and the waveforms acquired on oscilloscope are presented in the following figures. As can be seen from Figure 6 the sinusoidal current signal creates a sinusoidal magnetic field signal (both of them are in phase) what causes oscillations. The acoustic emission signals are rapidly changing when current and magnetic field are crossing the zero value.

From the waveforms obtained it can be deduced that the signal detected by the WSα sensor depends on the magnetic field strength. In the case shown, the magnetic field decreased with the distance. When the distance between the sensor and the center of the coil was 85 mm, there was no noise interference in the signal obtained and the magnetic field was not observable.

The next tests covered AE sensor response to rectified direct current flow. Again, the same coil was used, and the power diode was placed in series connection. Similar to previous tests the magnetic field values were recorded along with the sensor—coil increasing distance. In this case, no specific parasitic signal of significant value was detected, so it can be concluded that the rectified current and the constant sign magnetic field have no impact on the AE signal readings (see Figure 7). The presented results depicting behavior of AE transducer are valid for strong magnetic fields produced intentionally with means of current flowing across massive inductor. It should be noted that in practical application, there will not be such high values of magnetic fields generated by semiconductor structures.

These relatively weak fields will occur mainly due to electrons vorticity transport [28] and most of it will be “intercepted” and dissipated in metal plates which are integral part of a press-pack GTO casing.

### 2.2. AE signals Detection of GTO in Rectifying Mode of Operation

In order to get acoustic emission signals coming out of brand-new and unused thyristor in rectifying mode the laboratory test stand was prepared. Because of the nature of acoustic emissions, the longitudinal and shear wave propagation in solid materials comes mainly from rapid movement of the material particles during cracks on a micro scale. Much bigger dislocations, cracks and fractures can be the source of low frequency signals which in turn can be detected with use of electromagnetic field detector [29,30]. The latter method was not considered in the accomplished tests but due to the necessity minimizing the influence of the magnetic fields on the readings from the transducer all elements of the investigated system were placed as far apart as possible. Similar to previous tests the Hall effect magnetic field probe along with WSα was used. Both sensors were placed on the top of metal case on the anode side just like in Figure 8.

The GTOs firing circuit was attached to 30 V, 20 A DC supply which provided enough current for the gating circuitry. Unlike most of classic SCRs used in experiments a GTO needs firing current all the time when in conducting mode and its value heavily depends on the conducted current.

The thyristor and AE transducer were pressed down firmly with a non-conductive acrylic plate. To improve wave propagation into the transducer the contact surface was coated with a silicone-based gel. The thyristor anode-cathode terminals were supplied with alternating high-current, low voltage supply.

The tests carried out consisted in supplying the thyristor anode circuit with controlled by high-current autotransformer alternating voltage and firing the gate circuit by applying DC current to the gate-anode terminals. This forced the flow of rectified current of values dependent on the applied voltage, leads and the semiconductor structure anode-cathode resistance. The tests were conducted for 40, 60, 80 and 100 Amperes, respectively. Chosen results are presented in Figure 9.

As it can be observed the magnitude of acoustic emission signals (raw data) increased with increasing current values. Because the tested thyristor is capable of long-term conducting 1200 A (with proper cooling) the magnitude of raw AE signal may reach roughly 80 mV (assuming a linear increase with current—Figure 10) so in order to get the full spectrum of acoustic emission signal the proper type of transducer should be used.

The oscilloscope with connected to the AE transducer, Hall effect sensor and a Dietz ammeter recorded three waveforms that were later imported into the Matlab workspace in order to perform offline signal analysis.

The last, additional test that was performed covered pencil lead break. The pencil lead has been broken on the flat, hard surface of the upper pressing plate. The objective of such test was to check if it was possible to get external emission signal in the presence of current flow and line thyristor commutation in rectifying mode. As it can be seen in Figure 11, the amplitude of the pencil break signal was so high in comparison to regular signals coming from inside of conducting thyristor the vertical scale of the scope was increased 5-fold from 2 mV/div up to 10 mV/div.

While conducting the test the broken part of the lead bounced on the surface of pressing plate thus additional emission signal occurred. The data obtained in the test were transferred into the Matlab and FFT analysis of the signals was performed.

## 3. Results

For the signals obtained in experimental studies in the time domain, FFT conversion was used in the frequency range of acoustic emission extended up to 5 MHz. As it turned out the only visible bins of spectrum are present in a very low frequencies range as can be seen in the Figure 12 (blue, vertical line overlapping AE signal axis).

This clearly shows that the inner structure of the GTO thyristor during regular rectifying operation is not a source of high frequency acoustic signals. The obtained results of the fast Fourier transform were magnified, and the frequency axis limited up to 2 kHz.

The magnitude of transducer signals increases with the current amplitude and the significant spectrum bins are observable up to 600 Hz. From the Figure 13 it is clear that the elastic wave frequencies inside the semiconductor structure depend only on frequency of rectified current. The displacement that takes place in the material of press-pack creates signal of base frequency (50 Hz) and wave propagates in all directions of the thyristor volume. On the basis of the spectrum analysis, it can be seen that the harmonics present in the signal are multiples of the fundamental 50 Hz harmonic. They are generated by reflecting elastic waves from the boundaries of the thyristor structure. Higher values of the signal are sensed by AE transducer located closer to the reflected waves and main source of material displacement and friction.

As it can be observed in the Figure 14 the transducer detected the pencil break elastic waves, but the energy of the signal overlapped signals coming from the internal material expansion and all additional reflected waves were present inside the GTO structure.

The FFT analysis seems to be good enough for signals of short duration analysis (up to a few cycles) but it does not take account of thermal expansion which shows up in long time operation.

In order to get the desired information about how acoustic emission signals are changing with temperature deviations the system performing the power spectral density in real-time should be chosen.

## 4. Discussion on Results

The following conclusions can be drawn from the carried out experimental tests. It is possible to use a broadband AE sensor to determine the acoustic emission signals of a brand new GTO thyristor operating in rectifier mode. The obtained waveforms are characterized by a strong dependence on the thyristor switching frequency. Regular systems equipped with preamplifier filter out low frequencies but in the case of semiconductors this information may be useful for on-line condition monitoring or diagnostics. According to the nature of flexible wave propagation, the AE sensor also detected waves reflected inside the semiconductor structure which could be used to observe changes in volume caused by abnormal conditions such as overheating, internal dislocations or a decrease of wafer plate pressure.

The conducted tests did not show the existence of signals of frequencies typical for the acoustic emission band (range from 1 kHz to 1 MHz) during regular GTO operation. The WSα sensor is sufficiently resistant to electromagnetic interference occurring on press-pack enclosure metal plates and, as the test results have shown, the magnetic fields resulting from a direct current flow up to 100 A do not interfere with the AE sensor’s operation. According to our tests the use of such type of transducer in the vicinity of inductive elements generating strong magnetic fields will result in creation of parasitic signals, which may cover useful acoustic emission signals.

## 5. Conclusions

The presented method can be considered as a new approach to diagnose press-packed power electronics devices. As of now there is no widely available information about practical applications of acoustic emission signals and their changes with degradation, aging or any other concerning processes occurring in switching semiconducting devices. The obtained AE signals can be used as a reference in online monitoring systems which supervise the operation of the thyristor. By comparing them to the actual signals coming from a working GTO any changes noticed can be a sign of disturbances occurring in the structure. Because of possible internal semiconducting structure dislocations, cracks and fractures high frequency AE signals are expected to happen although up to now no evidences of such an effect were presented. On the other hand, due to multilayer structure aging observable, especially after prolonged time, the signal characteristics in current amplitudes coming especially from gate circuitry structures, slight but notable changes are expected to show up. The obtained results will be then some kind of a benchmark taken for good, fully operational thyristor so any signals which does not fit the pattern may be analyzed as a possible structure degradation.

In order to fully check the suitability of the presented method for testing the state of a semiconductor system, additional tests should be carried out, including prolonged exposure to higher values of currents (at least up to the nominal values) and power supply of the thyristor with the nominal medium voltage. The latter requirement makes it necessary, in order to ensure the safety of the equipment and operators, to use insulating spacers (e.g., mica plates), which will act as an electrical insulation but disperse a large amount of useful high frequency acoustic emission signals. This makes it necessary to use other types of sensors prone to strong electrical field for example made out of fiberglass.

## Figures and Tables

**Figure 1 sensors-21-00070-f001:**
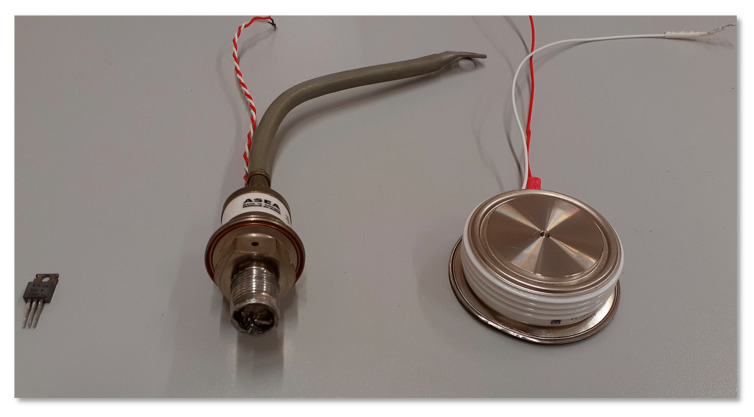
Example of thyristors in plastic, stud-mount and press-pack packaging.

**Figure 2 sensors-21-00070-f002:**
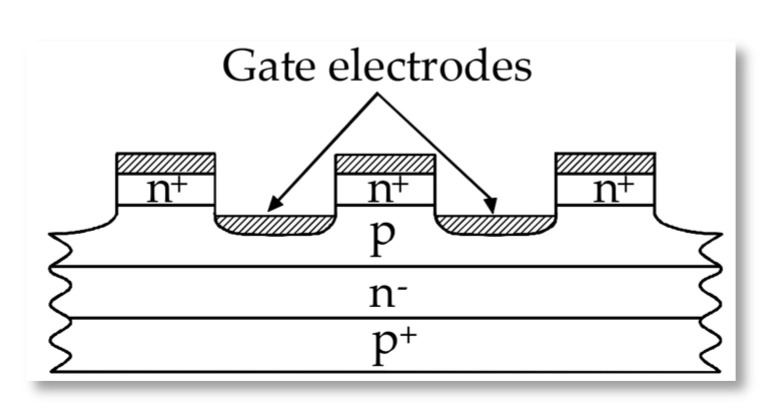
Cross section of a GTO showing the cathode islands and interdigitation with the gate (p-base).

**Figure 3 sensors-21-00070-f003:**
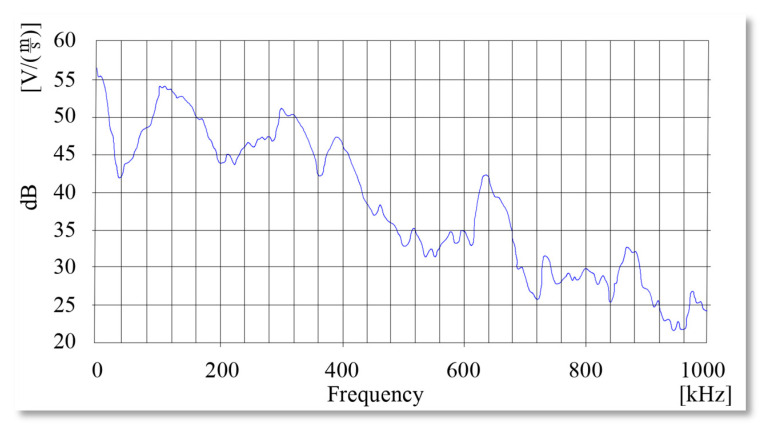
WSα sensor signals amplitude attenuation dependent on the frequency.

**Figure 4 sensors-21-00070-f004:**
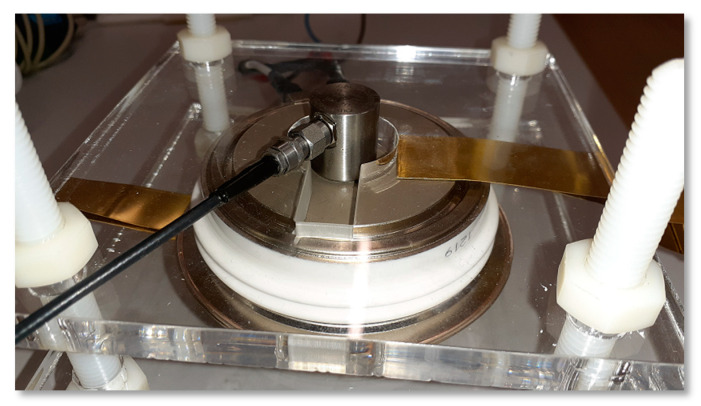
The acoustic emission sensor in the metal case placed on the GTO.

**Figure 5 sensors-21-00070-f005:**
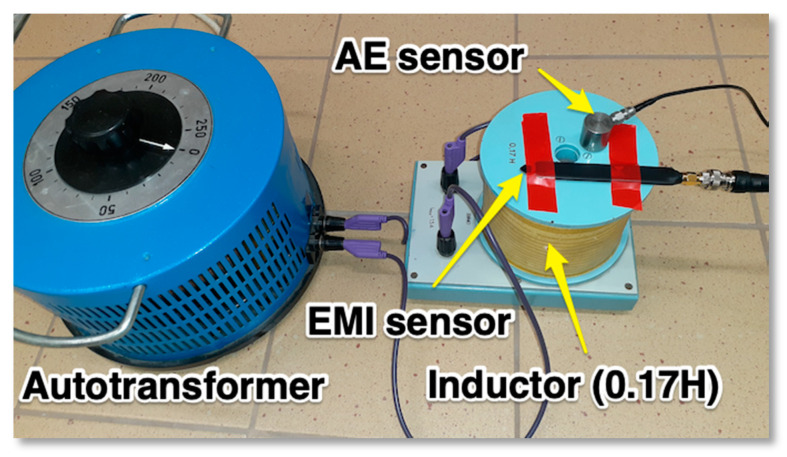
The sensors for AE and magnetic field signals detection.

**Figure 6 sensors-21-00070-f006:**
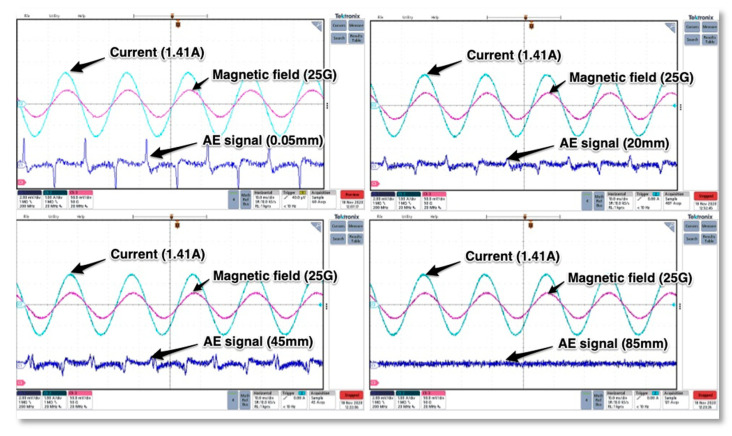
The magnetic field, current and AE signals obtained for different distances of the AE sensor from top of the inductor.

**Figure 7 sensors-21-00070-f007:**
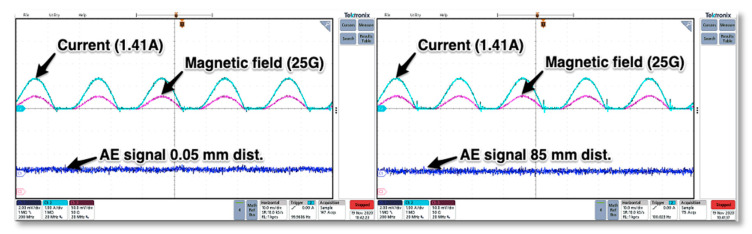
The test AE and magnetic field signals for rectified direct current flowing across the induction coil.

**Figure 8 sensors-21-00070-f008:**
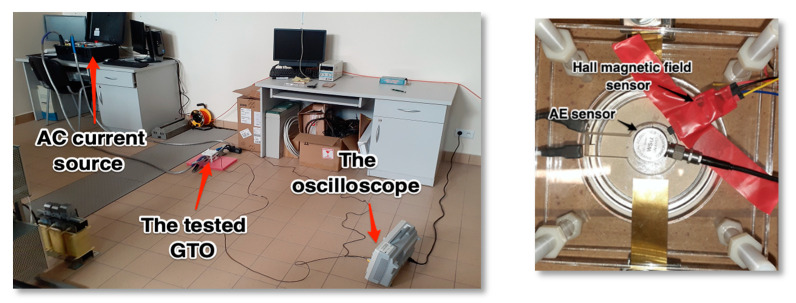
The laboratory test stand arrangement (**left**) and GTO 5SGS16H2500 symmetrical thyristor with AE and Hall effect sensors placed upon without a top pressing plate (**right**).

**Figure 9 sensors-21-00070-f009:**
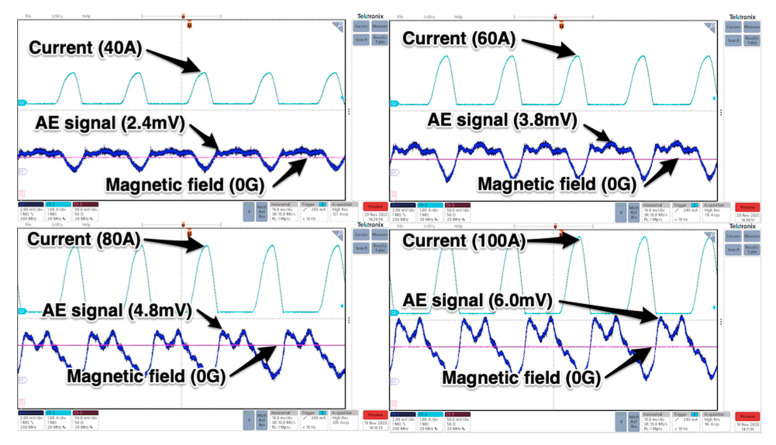
The chosen waveforms of raw AE signals (blue) detected by WSα transducer placed on the anode plate of conducting different currents (rectifying mode) GTO. Current and magnetic field waveforms placed for reference.

**Figure 10 sensors-21-00070-f010:**
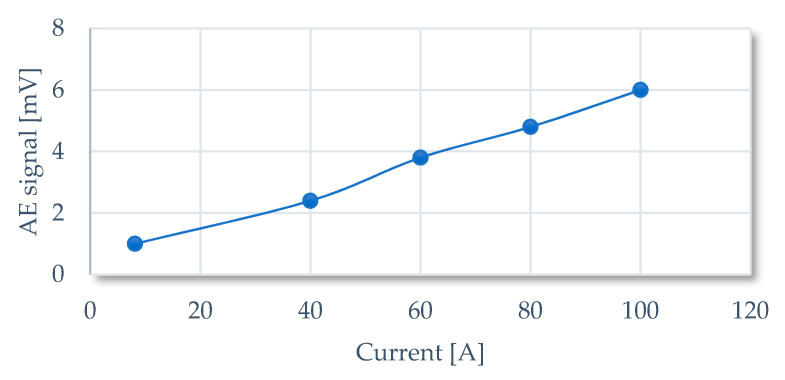
Graph showing the emission signal magnitude dependence as a function of the current flowing through GTO.

**Figure 11 sensors-21-00070-f011:**
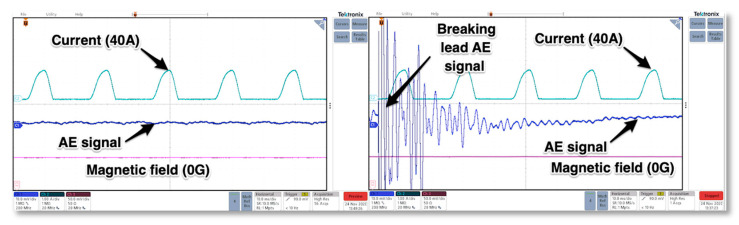
Waveforms of current, magnetic field and acoustic emission in regular GTO rectifying mode (**left**) and AE signal of break pencil lead test while GTO operating (**right**).

**Figure 12 sensors-21-00070-f012:**
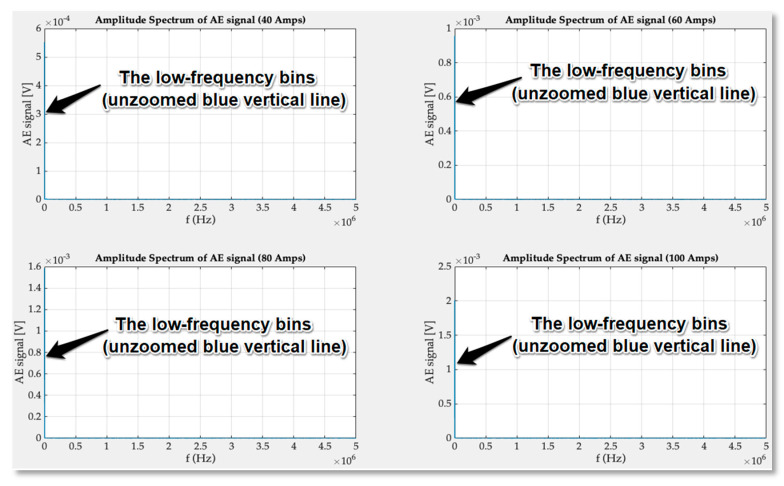
Wideband frequency spectrum of acoustic emission signal obtained for 40, 60, 80 and 100 amperes current.

**Figure 13 sensors-21-00070-f013:**
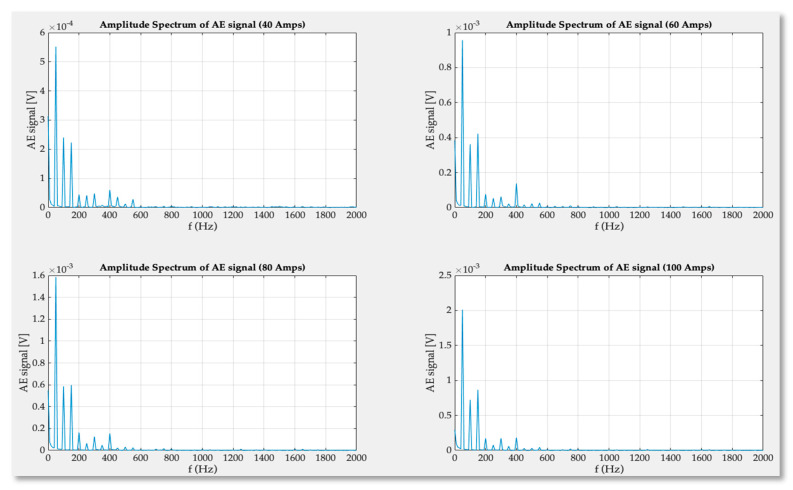
Narrowed frequency spectrum of signal recorded for 40, 60, 80 and 100 amperes anode-cathode rectified current.

**Figure 14 sensors-21-00070-f014:**
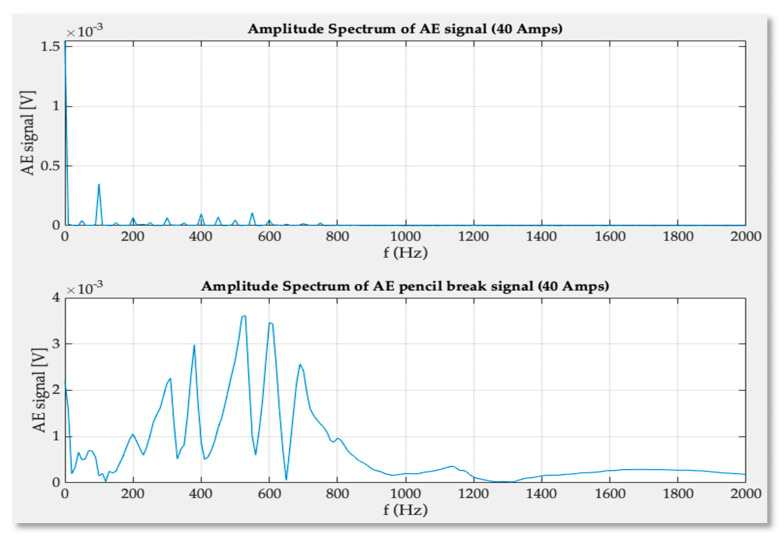
The frequency spectrum of signal recorded for 40 amperes anode-cathode rectified current (**upper**) and pencil lead break test effects in the presence of 40 A current (**lower**).

**Table 1 sensors-21-00070-t001:** Coefficient of thermal expansion for chosen materials.

Material	CTE (mμm/mK) at 300 K
Silicon	4.1
Copper (baseplate and pole pieces)	16–16.7
Al_2_O_3_ (Aluminum Oxide AL98)	6.2
Tungsten (W)	4.5
Molybdenum (Mo)	4.9
Aluminum (Al)	13.1
60/40 solder (Pb/Sn eutectic)	25

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
