# Peer review of "Identification of Gate Turn-Off Thyristor Switching Patterns Using Acoustic Emission Sensors"

_sensors, 2020, doi:10.3390/s21010070_

Round 1

Reviewer 1 Report

The article is very interesting and the subject matter deals with a very important problem that occurs on ships. The results obtained for different values of current provide the basis for continuous monitoring of GTO elements operation. There was no information in the article about the operation of this element in conditions close to failure. Such signal analysis could show what signal we obtain for critical conditions. It is probably not possible to easily achieve such working conditions, therefore the reviewer does not require adding these results. However, if the authors want to get rid of electro-magnetic interference, it is worth using fiber optic sensors.

Author Response

Hello there,

Thank you very much for so positive feedback of our paper. We are getting deeply into subject matter and I suppose to get very interesting results soon.

On behalf of all authors with kind regards

Maciek Kozak

Reviewer 2 Report

  1. The figure 12 is not clear.
  2. The sensors for AE and magnetic field signals detection in the paper should be calibrated or marked the type.

Author Response

Good day,

Thank you for positive feedback on the article and of course for the remarks.

Indeed, in the fist version figure 12 seems to be "empty", but if you take a closer look at the signal axis you will note a blue, vertical line which is depiction of low frequency FFT bins. In the present, altered version I've added information in the body of picture which you may find in attachement. It is self-explanatory now. The reason of including this picture was to show that in AE frequency range (100 kHz up to 1 MHz) there are no signals comin from structure, so in the case of any internal material disslocations acomplishing AE signals detection should be a straightforward process. 

As of AE sensor it has a factory calibration certificate which I have included in attachement. I don't think that this certificate is that crucial to show in the text body so I've add a little mention about a transducer calibration certificate.

The magnetic field Hall-effect sensor in this application should be considered more like an magnetic field indicator which was needed to check the approximate value of magnetic fied crossing AE sensor. We wanted to show how prone the AE transducer to magnetic field is and for presentation purposes only we have used Hall sensor. As a matter of fact this sensor has datasheet prepared by manufacturer which covers in very general way charasteristic of voltage vs. magnetic field but as is it has no calibration certificate. The readings are heavily dependent on the power supply voltage and its accuracy so the main issue was proper supply adjust.

Right now I am preparing answer to second reviewer so finished version of the article will be available in two days.

With kind regards

Maciek Kozak

Reviewer 3 Report

The paper presents an acoustic signal analysis methodology for gate-turn of thyristors. The paper contains a deep review and a lot of references to highlight the importance of the topic. However, some additional references and sentences should be added to highlight the usage of the acoustic analysis methodologies in power electronics or in other connecting fields to support better the novelty of the paper. Is it the first application of this methodology of this methodology in this field?

After reading the paper it is not clear for me, how is the first sentence of the abstract connects to the paper, I mean why is it so important or how is it connects to the novelty/motivation of the paper that these gto thyristors used in  a ship?

The set-up of the test equipment is described in details, but this section uses very limited number of references, such a strong statements, like "The only way to get to know the acoustic emission signals" should be supported by some references.

Figure 12 should be improved, it looks like empty.

Please clearify, that how is the different fatigue/aging phenomenas shuold be predicted by these tests, are the thyristors aged somehow before the tests?

There should be a separate conclusions section at the end of the paper.

Author Response

Good day,

First of all I would like to appreciate positive feedback to subject matter. Thank you.

As a matter of fact the literature's contribution to area of acoustic signals in thyristors and power electronic devices in general is not that great (most interesting, "pioneering" articles are cited in the text body). After digging the net there is not much to add. We as a team are developing (from early 2015) similar diagnostic system devoted mainly to detect AE signals in IGBT transistors and there are some contributions mainly from Finland, China and Japan. So yes, I may say that presented method is new (or nobody presented and described it yet clearly) regarding to other types of diagnostic methods used widely in modern power electronics.

All of authors are employees of Maritime University of Szczecin so it is natural that we are concerning about the ships implementation. Many new buildings have installed electrical drives as main propulsion and depending on the power the best idea is to use power electronics converters to drive them. In the case of high power systems there are a lot of thyristor converters present (esp. GTOs in cyclos and synchroconverters) but newest power transistors are coming to play. In the term of reliability the weakest link is the converter itself, also because the diagnostic methods leading to the detection of early structural damage are not perfect and when electrical parameters are changing due to damages its often too late for changeover propulsion drives or to perform safely shutdown procedures. We think that presented early-stage problem detection method (which is in development stage thou) will give more time to power management system and there will no blackout occur.

The statement about getting knowledge of acoustic signals was "soften" in altered version, but still the best way to get familiar with AE signals is the installation of AE sensor and get real-time unfiltered readings. I have also included two more of literature positions referring mainly to AE signals (of high and low frequencies) occurring in solid materials along with electromagnetic emission (EME) which can be detected with use of electromagnetic emission antenna.

Yes, the figure 12 looked somewhat empty, so I just added few words of explanation and in the updated picture there are arrows pointing to the vertical, blue lines (overlapping the axis) which are depicting low frequency bins. Intention was to present wide range of AE frequencies occurring while rectifying mode of operation. Magnified picture of the same data are presented in figs 13.

In the conclusion subsection I have added there is explanation that we are looking short high frequency pulse showing with internal disslocations/cracks/fractures due to imperfections of material, CTE and stresses. Aging (and wearing due to operation) of multilayer structures is the long-term, subtly process which mainly cause changes in semiconductor adjacent areas. Because of increased resistance we are suppose that this will affect current amplitude and maybe it will have impact on the current shape. The tested GTO was brand-new and unused so it allowed to get the "clean", unaffected signal which we consider as a kind of benchmark. The AE transducer was factory calibrated and has a certificate of calibration.

As I stated before we are into the subject and now we are preparing system to medium voltage tests, hoping to get similar results (with no electric field interferences).

The few typos and language errors were corrected.

Hope my somehow long explanations would help.

With kind regards 

Maciek Kozak

Round 2

Reviewer 3 Report

The authors answered all of my questions.